# Recovery of Rare Earth Elements from Wastewater Towards a Circular Economy

**DOI:** 10.3390/molecules24061005

**Published:** 2019-03-13

**Authors:** Óscar Barros, Lara Costa, Filomena Costa, Ana Lago, Verónica Rocha, Ziva Vipotnik, Bruna Silva, Teresa Tavares

**Affiliations:** Centre of Biological Engineering, University of Minho, Campus de Gualtar 4710-057, Braga, Portugal; laracardcosta@gmail.com (L.C.); filomenacpcosta@ceb.uminho.pt (F.C.); lagoanaelisa@gmail.com (A.L.); rocha.veronicapm@gmail.com (V.R.); ziva.vipotnik@gmail.com (Z.V.); bsilva@deb.uminho.pt (B.S.); ttavares@deb.uminho.pt (T.T.)

**Keywords:** *Bacillus cereus*, zeolite, biosorption, adsorption, rare earth element

## Abstract

The use of rare earth elements is a growing trend in diverse industrial activities, leading to the need for eco-friendly approaches to their efficient recovery and reuse. The aim of this work is the development of an environmentally friendly and competitive technology for the recovery of those elements from wastewater. Kinetic and equilibria batch assays were performed with zeolite, with and without bacterial biofilm, to entrap rare earth ions from aqueous solution. Continuous assays were also performed in column setups. Over 90% removal of lanthanum and cerium was achieved using zeolite as sorbent, with and without biofilm, decreasing to 70% and 80%, respectively, when suspended *Bacillus cereus* was used. Desorption from the zeolite reached over 60%, regardless of the tested conditions. When in continuous flow in columns, the removal yield was similar for all of the rare earth elements tested. Lanthanum and cerium were the elements most easily removed by all tested sorbents when tested in single- or multi-solute solutions, in batch and column assays. Rare earth removal from wastewater in open setups is possible, as well as their recovery by desorption processes, allowing a continuous mode of operation.

## 1. Introduction

The evolution of industrial activity over the last two centuries has led to various problems, such as an increasing need for consumption, accumulated pollution, and foreseen climate change. In parallel, this evolution has enabled the development of new technologies to tackle the problems raised [1,2]. Most transformation and production processes imply a resource deficit, as is the case with rare earth elements (REE) and precious metals [2]. Over time, the demand and need for these resources has grown exponentially, and in consequence, their traditional supplies are running out. Unlike for rock oil, there are no bio-derived alternatives for these elements, and their dispersion in the environment makes their recovery costly and difficult [1]. The rare earth group consists of seventeen elements (the lanthanide group of the periodic table, along with scandium and yttrium) and is divided into three categories: the light (La, Ce, Pr and Nd), the medium (Sm, Eu and Gd), and the heavy rare earths (rest of the lanthanide group elements and Y) [3,4]. REE contribute to important production sectors, including in the mundane (fluorescent lamps), high-tech (batteries, lasers, super-magnets) and futuristic (high-temperature superconductivity, information storage, conservation and transport of energy) fields, due to their diverse chemical, electrical, metallurgical, magnetic, optical and catalytic properties [5,6,7]. Currently, countries like China, the United States of America and Australia are the leading producers of REE, with China accounting for 90% of the global worldwide production [8]. These elements are difficult to eliminate from the environment, as they are non-biodegradable and their impact is reinforced by their accumulation throughout food chain [7]. At this point in time, REE are not yet recycled, although they are gaining attention due to their critical importance in high-tech products [1] and, in association with their increased demand, their purification and pre-concentration have been evaluated [9,10]. There are several methods to separate, pre-concentrate and purify these ions, including adsorption, chemical precipitation, and ion exchange [11].

Adsorption is recognized as being one of the most popular methods due to its simplicity, high efficiency and wide range of availability of sorbents [5,7]. Zeolites, widely used as adsorbents, are aluminum silicates with microporous structures [12], and they present greater chemical selectivity compared to silica and active carbon [12]. The structures of zeolites consist of three-dimensional frameworks of SiO_4_ and AlO_4_ tetrahedra, in which the replacement of the Si^4+^ by Al^3+^ produces a negative charge in the matrix [13]. This negative charge results in a strong affinity for transition metal cations and low affinity for anions and non-polar organic molecules, although this may be balanced by the exchangeable cations like sodium, potassium or calcium [13,14]. The ion exchange ability of zeolites allows them to incorporate different cations from solution. The pore size modulates the use of zeolites as catalysts [15,16].

Biosorption is a specific method for removing metal from wastewater [17,18], whereby certain types of biomass (usually bacteria, fungi, yeasts or seaweeds) bind and concentrate metal ions from aqueous solutions [19]. This mechanism presents some advantages over chemical ones, such as lower operation costs, reduction of required chemicals and deposition of produced sludge, and higher efficiency in the detoxification of diluted effluents [20]. Biological processes using non-living biomass arose as an alternative approach due to the fact that metal ions can be retained by biomass through surface precipitation, van der Waals forces, ion-exchange reactions, electrostatic interactions, complexation [21,22,23] or by a combinations of several of these processes [24]. On the other hand, metabolic pathways seem to play an important role when living organisms are used. Moreover, it is well known that the surface functional groups of microbial cells such as carboxyl, hydroxyl, amine, phosphate and sulfhydryl groups are involved in ion entrapment [25,26]. The recovery of REE retained by biosorption processes, using either batch or continuous mode, was reported in [27]. After the leaching of the elements from the support, they can be separated and purified through extraction using specific solvents, followed by selective precipitation. Nevertheless, the use of organic solvents is difficult to achieve sustainably, necessitating deeper research on this second step of the recovery.

The developed method is to be applied to real industrial and mining effluent in the near future, although some authors are already testing their work in the context of real scenarios [28], aiming to achieve economic and environmental benefits. The main objective of this work is the development of an environmentally friendly technology for the treatment of wastewater containing REE (lanthanum, La; cerium, Ce; yttrium, Y, terbium, Tb; praseodymium, Pr and europium, Eu), from a circular economy perspective, so that these elements can be reused in the production cycle. To achieve this, rare earth elements in aqueous solutions will be adsorbed onto zeolite and by a *Bacillus cereus* biofilm supported by the zeolite, and the two matrices will be compared. 

## 2. Results and Discussion

### 2.1. Effect of the Different REE on the Bacteria Growth

In Figure 1, the biomass concentration profile is represented at different time moments for an initial concentration of 20 mg/L for La, Ce, Y, Pr, Tb and Eu for single- or multi-component assays. This makes it possible to assess the xenobiotic effect of each element on the bacteria *B. cereus*, and possibly to evaluate the sensitivity of the bacteria towards the tested elements. Biomass concentration was calculated using a calibration curve, relating optical density to biomass concentration. Typically, a growth curve includes four phases: lag, log, stationary and death phases [29].

Figure 1 shows that in the lag phase, there is no significant difference between the tested elements, indicating that the bacterial culture needs the same time to endure intracellular changes to adapt to each of the elements. In the log phase, the biomass concentration increases significantly, whereas in the first 7 h of growing, the behavior is quite similar for all the tested REE. For Pr and Tb elements, the log phase is shorter, reaching 0.9 g/L and 1.1 g/L of biomass concentration, respectively, after the first 10 h of growth, in contrast with the other elements, suggesting that Pr and Tb might have a stronger xenobiotic effect on the bacteria. Bacteria adapted better to Eu compared to the other elements, and after 14 h of growth, the maximum concentration obtained was 1.23 g/L in the presence of Eu and 1.20 g/L in the presence of La.

In order to assess the impact that different rare earth metals, in single- and multi-component solutions, may exert on the morphogenesis and on the growth of *B. cereus*, several samples from all the experiments conducted with this bacterium were collected over time, dried, subjected to Gram staining, and subsequently observed under the microscope (Olympus BX51), Figure 2. 

It is possible to observe, in all the images, the presence of central spores, dormant cells produced by a variety of bacilli species, a response to either starvation or exposure to detrimental conditions [30,31,32], as well as the presence of oval-shaped cells resulting from the transformation of rod-shaped cells accompanied by a reduction in size due to exposure to toxic and nutrient deprivation conditions. It is also possible to observe that the majority of the microbial culture has changed from Gram-positive staining to Gram-negative staining, also a response to the exposure of the microbial culture to unfavorable conditions, which can affect the cell wall structure [32], making it thinner, more fragile and more diffuse [33]. 

During the microscopic analysis, it was also observed that the samples corresponding to the exposure of *B. cereus* to Ce and Pr (Figure 2B and 2D) were those that showed signs of increased stress and morphological changes, namely the formation of a higher number of central spores, accompanied by the alteration of shape and the reduction of size of the cell, as well as damage to the cellular wall, promoting the staining of the vast majority of the cells as gram-negative. These results were expected, since REE toxicity usually decreases as the atomic number of REE increases, probably due to the higher stability and solubility of the heavier REE [34]. The stress observed for the cells exposed to Ce could be related to the fact that Ce compounds are used to stimulate cellular metabolism or to improve the contents of protein, glucose, pyruvate, and carbohydrates related to enzyme activities, thus leading to an increase in growth and survival rate [35] as a result of the development of spores, as is the case of the present work; thus, higher levels of biosorption for this element is expected.

### 2.2. Adsorption and Biosorption Assays

#### 2.2.1. Zeolite

As expected, the zeolite surface observed by SEM, Figure 3A, is mainly rough, promising a high surface area. pH affects the metal solubility and the sorbent surface charge, as protons can be adsorbed or released [30,31]; therefore, controlling the pH is determinant. The tested REE begin to precipitate as hydroxides at high pH values [32]: La from 7.82 to 8.10, Ce from 7.60 to 7.80, Pr from 7.35 to 7.89, Eu from 6.82 to 7.30, Tb around 7.20, and Y from 6.75 to 6.83 [36,37,38]. pHzpc tests were performed in order to establish the assays’ pH, and these should correspond to the pH value at which the net surface charge of the adsorbent is electrically neutral. At pH < pHzpc, the adsorbent surface becomes positively charged, while at pH > pHzpc, the adsorbent surface becomes negatively charged. The pHzpc for zeolite (Figure 3B) is 9.6, and for pH values above this, the REE will precipitate as hydroxides. According to literature, the optimal solubilization pH for each REE is: La from 5 to 7, Ce from 4 to 7, Pr at 5, Y at 7 and Eu at 4.5 [7]. Taking this into consideration, the chosen pH for sorption assays was 5.0, as the majority of the REE present a satisfying sorption yield at this pH, and an acetate buffer of 0.1 M was used in order to maintain a stable pH.

A textural characterization including specific surface area (S_BET_), total pore volume (V_total_), micropore volume (V_micro_), mesopore volume (V_meso_) and the average pore size for the zeolite was performed, and the results are presented in Table 1. Zeolite 13X is a microporous solid, as the micropore volume corresponds to more than 70 % of the total pore volume. 

The adsorption capacity for each of the six REE was tested through single- and multi-component batch assays (Figure 4). The obtained results are presented with normalized concentrations, C/C_0_, over time.

Focusing on the single-component assays, a rapid initial uptake is observed for three REE (La, Ce and Pr), while the other three (Y, Tb and Eu) presented a slower initial uptake. Only two of the three REE with rapid initial uptake were completely removed from the solution, i.e., La and Ce. After 72 h of assay, there was still around 25% of the initial concentration of Eu in the solution, suggesting a slow continuing reduction of its concentration over time. Y, after 36 h assay, still presented around 35% of its initial concentration, meaning that it may need more time either to reach equilibrium or to be completely adsorbed. The results for Tb suggest that it reached a steady state after 24 h, which continued until the end of the assay. 

In the multi-component assays, there was a substantially higher initial affinity for La, Ce and Pr in comparison with each individual assays, exhibiting total removal from the solution in these new circumstances. The other three sorbates (Y, Tb and Eu) had an incomplete removal from the solution, where 30% to 40% of initial REE concentrations was still present in the solution, just as happened for the single-component assay (Figure 4). These three sorbates exhibit a continuous reduction of their concentration over time, suggesting a slower removal pathway. It is then considered that La has the highest affinity to the zeolite surface of all the tested REE, followed by Ce and Pr, since these achieved total removal from the solution, just as occurred in the single assays.

It can be noted in Figure 4 that all REE are affected by competition with other metals in the multi-component system, except for Ce, whose removal yield remains the same, while the removal of La and Pr is slightly affected by the presence of the other REE. The removal of Tb reaches equilibrium in 24 h when in a single solution, which was not observed in the multi-component assay. Also, Y removal in the multi-component assay was similar to that described for Tb, as the residual metal concentration after 36 h assay was delayed to 48 h when comparing single-solute with multi-solute systems. There is a small increase in the amount of Eu in the solution when reaching equilibrium, from 28% to 38%, when adding other solutes to the single solution, highlighting the competitive behavior between these metals for the sorbent surface. Y, Tb and Eu seem to have a lower affinity for the zeolite when compared with the other elements that are completely removed.

The pH was monitored during the single- and the multi-component assays, in order to avoid metal precipitation. As expected, the pH varied between 4.75 and 5.50, Appendix A. 

For practical purposes, the determination of the final uptake is relevant, both in single- and in multi-component assays, as displayed in Table 2 for all the sorbates.

La, Y and Pr exhibit an increase from single- to multi-component conditions, indicating a synergetic effect between sorbates, while Ce, Tb and Eu show a decrease in uptake, although Ce suffers a very minimal alteration. The increase in the uptake of Y in the multi-component assay is related to the fact that one of the assays did not reach equilibrium. The uptake reduction of Eu and Tb is a result of the competition between the sorbates.

Part of these observations may be justified by the ionic radius, which establishes the following order for the used REE, from the biggest to the smallest: La (250 pm) > Ce (248 pm) > Pr (247 pm) > Y (240 pm) = Eu (240 pm) > Tb (237 pm). The elements with higher ionic radius are those that showed better adsorption, following the same order as ionic radius. In fact, a higher ionic radius means a smaller hydration capacity, resulting in a weaker binding between the ion and the water phase, and allowing a higher adsorption, as has been reported in the literature for heavy metals [39,40]. Additionally, the zeolite pore size, 20 Å (Table 1) is big enough to allow REE entrapment without the pore closing. The sequence of the molecular weight of the REE is Y (88.905) < La (138.905) < Ce (140.116) < Pr (140.907) < Eu (151.964) < Tb (158.925), correlating with the REE’s adsorption, those with lower MW have higher adsorption yields, except for Y. In the case of heavy metals, it is reported that MW and sorption are directly correlated [40], while for the REE, the MW and sorption are indirectly correlated. 

Zeolite samples used in single- or multi-component adsorption assays were analyzed in SEM/EDS to assess the presence of the sorbates within the zeolite. A low metal concentration in the zeolite is expected, as for each assay the proportion between zeolite and each REE is 250:1 (2000 mg of zeolite per 8 mg of REE, assuming a total REE removal). The analyses detected REE elements in very small concentrations when compared with the components of the zeolitic structure. It is important to mention that values lower than 2% should be considered to be forced and/or background influence. To confirm the entrapment ability displayed by zeolite, SEM/EDS was performed on the surface of a zeolite used with sorbate concentrations higher than 20 mg/L, for a 72 h assay. The most abundant elements detected were oxygen (O), aluminum (Al) and silicon (S), which was already expected, since these structures are alumina silicates. Besides those elements, there were some ions, such as Na^+^, Mg^2+^, Ca^2+^, coming from the distillated water used to stabilize the charge deficit on the zeolite surface. The results are displayed in Table 3. 

This analysis confirms that the REE are captured within the zeolitic structure, as the proportion between the zeolite and the REE is much lower than 250:1, allowing a better detection of the REE entrapped within the zeolite. Eu, La and Pr were detected to a higher degree than the rest of the elements, which is in agreement with the adsorption assays, since these were the elements that showed the highest adsorption yields.

#### 2.2.2. Bacteria in Suspension as Sorbent

The sorption process by bacteria is highly dependent on the morphology and composition of the surface of the microorganism, and more specifically on membrane proteins such amines, amides and alkynes, which showed the ability to sorb REE and then to decrease the concentrations of these elements in solution, meaning that the concentration of biomass in the system is a crucial factor to be considered and optimized in this type of study. Another relevant factor is the measurement and control of pH during the assays, since the entrapment of REE from solution is highly dependent on pH, due to the direct influence of the pH on the bacteria surface groups. Low pH can lead to protonation of groups such as carboxyl and ketones, meaning that the binding sites are occupied; and high pH can lead to deprotonation of those groups, leading to a negatively charged biomass surface [41]. These parameters can affect the available number of active sites on the surface, and in consequence the adsorbent–adsorbate linkage. For this reason, all experiments were performed with 0,1 M acetate buffer at pH 5. 

Batch experiments with biomass were performed in order to evaluate the capacity of *B. cereus* to retain REE. The results are shown in Figure 5, with normalized concentrations for single- and multi-component assays over time.

Single-component assays started with a pre-established concentration of 20 mg/L for each element and with a fixed concentration of biomass of 0.5 g/L, reaching a final removal of 80% or higher for all elements. There was a rapid uptake of each sorbate within the first 10 h, with a special focus on Eu and Ce, which had a slightly higher uptake in comparison to the other elements, apart from the fact that the biosorption yield was similar for the six elements under study. Again, this relative behavior is related to the ionic radius of the elements, since Ce is one of the elements that has a higher ionic radius, as mentioned above. Biosorption data is also in agreement with REE relative toxicity. Removal percentage diminishes with the xenofobicity of the element (Ce > Tb > Eu > Pr > La > Y). The higher biosorption percentage measured for Ce is related to the fact that Ce compounds are used to stimulate cellular metabolism and enzymatic activity, promoting cell growth and resilience as a result of spore development [35]. 

After 48 h assay, it was not possible to achieve an acceptable REE removal in the multi-component system, and a concentration of biomass of 0.5 g/L; Figure 5. The maximum REE removal was around 20%, a much lower value when compared to the single-component assays, for all REE. The multi-component assays also demonstrated the lack of selectivity of the biomass towards the tested sorbates. The monitoring of the pH at the start and at the end of the assays confirms that there was no precipitation of REE; Appendix A.

Considering the total uptake, in terms of the ratio between the mass of sorbate and the biomass, achieved in the multi-component solution assays compared to individual experiments, Table 4, it is evident that the biomass was not selective towards any element in the study, as uptake values were of the same order of magnitude, and they were quite similar. The presence of other sorbates significantly reduced the uptake results, which can be explained by the saturation of the active sites of the bacteria and the competitive effect between sorbates. 

It has therefore been established that, for single-solute assays, zeolite presents higher uptake values for all the REE when compared to *B. cereus*, revealing a higher affinity. Nevertheless, it is important to take into consideration that the zeolite assays were performed with a zeolite concentration of 5 g/L, while biomass was tested with only 0.5 g/L. This 10-fold difference may justify the difference in REE removal; in addition to the concentration, the surface area of zeolite may justify this difference; Table 1. In the multi-component assay, *B. cereus* showed no affinity for any REE, in contrast to zeolite, which had a higher affinity for La, Ce and Pr. Bacteria showed a higher uptake for all metals, which could be explained by the lower concentration used when compared to the zeolite. 

#### 2.2.3. Supported Biomass on Zeolite

Foreseeing a practical application of these matrices for REE recovery, batch assays were performed with *B. cereus* biofilm supported on zeolite, so that some advantage might be gained from an eventual synergy between the two materials. Biofilm structures show tolerance towards changes in environmental conditions such as nutrient deprivation, predation, exposure to toxic chemicals such as pollutants in high concentrations, or other environmental stress factors such as, for instance, changes in pH, temperature, salt concentration and water content. These characteristics of biofilms are relevant in the bio-rehabilitation of contaminated water, and the understanding of these phenomena will help to improve and develop strategies for bioremediation [42].

Figure 6 shows the SEM images confirming the presence of *B. cereus* biofilm on the zeolite surface (see arrows). Although all tests were performed in the very same conditions, to allow biofilm formation, the percentage of biofilm formed when compared with the available surface area of zeolite is quite reduced, and this will obviously influence the adsorption efficiency of the matrix. 

The sorption data obtained with this combined material, for the tested conditions (multi-component solution of REE), are displayed in Figure 7. 

The results indicate higher removal efficiency for La, Ce and Pr when compared to the others at equilibrium. The biosorption accomplished at least 60% removal for all of the elements; and for La and Ce, the removal reached 90%. As happened with the adsorption experiments with zeolite for REE multi-component solution, biosorption was more efficient for some elements, following the order La > Ce > Pr > Y > Tb > Eu. It may be concluded that zeolite can achieve better removal efficiencies than the supported biofilm, but it is important to highlight that these processes are highly dependent on the surface area and on the number of active sites available, and the biofilm tends to reduce those parameters when covering the zeolite surface. Nevertheless, the zeolite-biofilm matrix is a stable and strong structure that can be used in the rehabilitation of contaminated water and certainly could be improved and adjusted for specific sorbate targets.

As before, the pH variation during the assays was minimal, circa 5.00, and the specificity of each sorbate behavior was mainly due to differences in ionic radius. La and Ce have the biggest ionic radii of the REE group, and they reached the highest removal efficiency. 

The uptake values presented in Table 5 also validate the higher affinity of La and Ce. The uptake values for Tb and Eu were very similar, revealing the possible competition between these elements, given that they have similar ionic radius (Tb—237 pm, Eu—240 pm) and molecular weight (Tb—159 g/mol, Eu—152 g/mol).

Overall, the supported biomass on zeolite showed better results when compared to those for suspended biomass in terms of C/C_0_ versus time (h), although the uptake values were smaller. This difference in uptake could be a result of the addition of zeolite, which increased the sorbent mass. When compared to zeolite, the supported biomass on zeolite had a smaller uptake, as well as lower sorption, which could be explained by a pore reduction due to the presence of the biofilm.

### 2.3. Adsorption and Biosorption Assays

The kinetics modelling of the data was performed using the non-linear equations for the Pseudo-first order (PFO) and Pseudo-second order (PSO) models, as the linear forms of these equations are transformations of the non-linear forms, which alter their error distribution and distort the fitting parameters [43,44,45].

Also, the non-linear equations of Langmuir and Freundlich isotherms were used for the equilibria modelling in order to avoid a change in the error structure, which is inherent to the commonly practiced linearization of the above models [46,47,48].

#### 2.3.1. Sorption Kinetics Modelling

PFO and PSO models were used to fit the kinetics results for the single-solute assays, and fitting parameters are presented in Table 6. The best model was selected according to the Akaike Information Criterion (AIC). AIC is an estimator of the likelihood of a model, the best being the one with the minimum AIC.

Considering the correlation (R^2^) and the AIC values, the PSO model fits better to the experimental data for both zeolite and suspended biomass, apart from La and Tb with suspended biomass, which had better fitting for PFO. The representation of the model fitting for zeolite and suspended biomass can be found in Appendix A, respectively. As PSO showed the best fitting, meaning that the adsorption follows a chemisorption mechanism, this may be explained by the interactions between the REE and the chemical groups of the bacteria wall. The same applies for the zeolite, although the interactions established by the REE will be with the oxygen atom, as its electronegativity will probably be responsible for the REE cations entrapment, helping the total charge stabilization of the zeolite (the zeolite is negatively charged as result of the replacement of silicon, Si^4+^, for aluminum atoms, Al^3+^). Overall, suspended biomass presents a higher adsorption capacity at equilibrium, q_e_, as well as a higher affinity constant, k, for all REE that had PSO as the best fitting model, except for Ce. When q_e_ and k for PFO are compared, the suspended biomass emerges with the best values, as before. These results suggest that *B. cereus* is the best solution as adsorbent for the REE entrapment under these conditions, as it presents the highest capacity at equilibrium, reducing the adsorbent mass needed, and the lowest affinity constant, resulting in a reduction in the time needed for adsorption.

The kinetics results for the multi-solute assays and fitting parameters are presented in Table 7 for zeolite, suspended biomass and supported biomass on zeolite.

As happened for the single-solute assays, the PSO model fits better to the experimental data for all tested adsorbents except for Pr with suspended biomass. The representation of the model fitting for zeolite, suspended biomass and supported biomass on zeolite may be found in Appendix A, respectively. When the affinity constants of supported biomass and suspended biomass are compared, it can be seen that the values of the first are lower than in the last two cases; however, they are higher than those determined for the zeolite. This difference could be a result of the reduction in the zeolite surface in contact with the solution due to coverage by the biofilm. This surface reduction leads to a reduction of possible binding sites, and therefore to reduced chemisorption.

As before, suspended biomass presents a higher q_e_, as well as a higher k, for all REE, but it is important to note that the uptake is normalized by the adsorbent mass, which is quite a bit lower for suspended biomass, and this could explain the higher q_e_ and k.

The comparison of the fitting plots between single- and multi-solute assays indicates that adsorption equilibrium for each sorbate, when in multi-solute, takes longer to be reached (fitting representations for zeolite and suspended biomass: see Appendix A for single-solute and Appendix A for multi-solute assays), probably due to the dynamic competition for adsorption sites, and this fact will be determinant in the up-scaling of the adsorption set-up. 

#### 2.3.2. Sorption Equilibria Modelling

The single REE adsorption isotherms for zeolite are presented in Table 8, and the graphs are presented in Appendix A.

The Langmuir model is the best description of the data for all REE, except for Ce. The Langmuir isotherm assumes that the binding sites of the adsorbent are equivalent, resulting in the formation of just one monolayer of adsorbed molecules. The affinity order for the REE, considering the calculated q_max_, is Y > La > Pr > Eu > Tb. The Freundlich isotherm model assumes a multilayer adsorption, with interactions between the adsorbed molecules. Adsorption of Ce is better fitted by Freundlich, with an n higher than 1 and lower than 10 [49], meaning that this adsorption is favorable.

Similar fittings were performed for further assays, starting with a multi-solute solution, for different sorbents (zeolite, suspended biomass and supported biomass on zeolite), the results of which are presented in Table 9, and the fitting graphs of which are presented in Appendix A, respectively.

Overall, the Langmuir fitting was better when zeolite was used as sorbent, assuming the formation of just one monolayer of adsorbed molecules, except for La and Pr. One thing is common to the two tested models, which is the order of three most adsorbed REE, La > Ce > Pr, suggesting that these ions have the highest affinity for the zeolite, with a predominance on the entrapment processes. The electronegativity potential is proportional to the affinity of that element to a specific biosorbent [50,51], and the order for the electronegativity (Pauling scale) for the REE is Y (1.22) > Eu (1.2) = Tb (1.2) > Pr (1.13) > Ce (1.12) > La (1.1), which is inverse to the order of the three most adsorbed REE, so other factors are influencing the whole process.

For the suspended biomass, the Freundlich isotherm fits the data more adequately (with the exception of Eu), showing lower AIC values and satisfying correlation values. These results suggest that there is a multilayer sorption for the bacteria. Based on the coefficient *K_F_*, the REE affinity followed the order: Pr > Tb > Ce > La > Y. Y is the element that takes the longest to reach equilibrium (Appendix A), with a lower value of q_e_ and lower affinity potential for the biomass. 

As happened for the zeolite, the Langmuir fittings were better for supported biomass on zeolite for all REE except La. The highest adsorption capacities, q_max_, were obtained for Y and Eu, with 10.2 mg/g and 9.01 mg/g, respectively, as both elements presented higher electronegativity of all tested REE, a characteristic reported to correlate with affinity to a specific biosorbent.

Comparing the three adsorbents, suspended biomass exhibited better fittings for the majority of the tested REE, with the exception of La, which showed better fitting for the zeolite, and Y, which showed better fitting for supported biomass.

### 2.4. Desorption

Desorption of REE is as important as sorption, since it allows the recovery of the sorbates and their re-use in new applications. The eluents to be used must be non-toxic, cause no damage to the sorbent, allow its reuse, and offer maximum recovery at the lowest possible concentration [52] and contact time. Nitric acid, HNO_3_, was selected as leaching agent due to its efficiency, which is acknowledged in the literature [53,54,55].

Desorption might also explain the affinity between REE and zeolite, considering that REE which do not have a high recovery may have a high affinity, and that the opposite could also happen. The results for the recovery are presented in Figure 8. 

The recovery of La presented little difference between acid concentrations and tested time. Ce had the worst recovery of all REE, with no evident differences between the tested conditions. Acid concentration had an initial influence on the recovery of Y and Tb, with the highest recoveries being registered for the higher acid concentration after 0.25 h and 1 h, respectively. Pr was the only REE with almost no difference in recovery between the different tested conditions. Eu recovery exhibited an initial influence of acid concentration, reaching the highest recovery with the smaller acid concentration. In general, and regardless the previous adsorption conditions, all REE had recovery values higher than 60%, apart from Ce, after around 1 h of assay. Higher acid concentration seems to work very well in the recovery of REE in a single-component solution.

Zeolite used in the multi-component assays was also used in desorption, with results presented in Figure 9.

The HNO_3_ concentration and the assay time had no effect on the recovery of La, Ce and Pr. The recovery of the other REE, namely Y, Tb and Eu, showed a similar trend, and while there were some differences in the beginning of the leaching process among the acid concentrations tested, the differences disappeared. It should be noted that the REE with the highest removal (Figure 5) were those with the smallest recoveries. The order of the REE in terms of highest adsorption was La > Ce > Pr, although this order was swapped for recuperation to Pr > Ce > La (with recoveries of 20% of La, 30% of Ce and 40% of Pr). The change in the order could be explained by the locations on the zeolite where the REE are adsorbed, which is agreement with the possible higher affinity between the REE and the zeolite.

There was an overall decrease of REE recovery for La, Pr and Y, while there was an increase for Ce, when starting with single sorbate adsorption compared to multi-sorbate adsorption. 

Overall, the recovery of REE from zeolite is a promising way to ensure the reuse of these REE, as 1 h seems to be enough to reach the highest recoveries for all REE. Nevertheless, some more tests should be performed with respect to acid damage to the zeolite during this treatment.

After the desorption step, all REE will be mixed, and it is necessary to add another step to obtain them individually. This could be performed by differentiated precipitation by exploring the pH or other salts that could allow the individual obtaining of REE; solvents are another option that could be considered for the extraction [56].

### 2.5. Column Assays

Column assays were carried out to simulate the recovery of REE from wastewater at an industrial scale. These assays were performed without acetic buffer, simulating a real effluent, although this may present some difficulties regarding pH. This was measured during the assay at its outflow in order to assess the possibility of REE precipitation. The pH of the feed solution was adjusted to 3, and then it was passed through columns containing zeolite, with or without *B. cereus* biofilm. Biofilm formation was confirmed by SEM images from samples of the zeolite within the columns; Figure 10.

The presence of bacteria on surface of the zeolite is confirmed, since the roughness of the zeolite (Figure 10B) is not evident to the same degree when the biofilm is present. The segments in green indicate the measurement of a *B. cereus* length within the biofilm, which is around 1.5 ± 0.33 μm (Figure 10A).

The REE concentration tested in this open system was reduced from 20 mg/L to 10 mg/L, although in most real systems, REE concentrations are expected to be even lower. The experimental data for the REE multi-component adsorption in the column are presented at Figure 11.

The biotoxicological assays revealed a stronger effect of Pr and Tb and a weaker effect of Ce, which makes sense, as the column assays with biofilm show a similar behavior in sorption of Pr and Tb and a better outcome for Ce. The biofilm does not enhance the sorption process during the first hour of the assay, as zeolite presents lower C/C_0_ values. The biofilm formation on the surface of the zeolite may be responsible for the slower sorption in the beginning, due to a possible reduction of the surface area available and increased mass transfer limitations. At 6 h of assay, the column with zeolite supported biofilm begins to present higher sorption than the column with zeolite only, which is probably due to the increased number of sorption sites on the biofilm surface, and could even be due to its metabolic activity. After almost a 30 h run, the biofilm adds no value to the sorbent.

As before, pH was measured during the whole assay (Appendix A), with values always under 6 when using zeolite. When biofilm is present, it starts with values higher than 6 during the first 9 h, then reduces along the assay. 

After the adsorption process, leaching was tested for sorbate recovery and concentration for future purification and re-use. The desorption assay was carried out with 0.1 M of HNO_3_, similar to the batch assays, and the results are presented in Figure 12.

Also, here the biofilm seems to offer additional resistance to REE recovery, suggesting that the sorbed metals should be further used within the zeolite, after biofilm destruction, such as in catalytic applications, rather than going through any leaching procedure, as tested [57]. Alternatively, the reuse of the zeolite after desorption in new cycles of REE removal could be considered, as has been reported for diverse other sorbents [58,59].

## 3. Materials and Methods

### 3.1. Rare Earth Elements

The batch and continuous assays were conducted with rare earth elements from stock solutions previously prepared with a concentration of 1000 mg/L, with the dissolution of each metal salt in distillated water. Europium, Eu, (EuCl3.6H2O; 99.9%) and cerium, Ce, (Ce(NO3)3.6H2O; 99.5%) were purchased from Acros Organics (Geel, Belgium). Lanthanum, La, (La(NO_3_)_3_.6H_2_O; 99.9%), praseodymium, Pr, (PrCl_3_.xH_2_O; 99.9%), terbium, Tb, (TbCl_3_.6H_2_O; 99.9%) and yttrium, Y, (YCl_3_.xH_2_O; 99.9%) were purchased from Alfa Aesar (Kandel, Germany). The multi-element ICP quality control standard solution, which includes 200 mg/L of each element under analysis, was purchased from CPAchem (Stare Zagore, Bulgaria).

### 3.2. Bacteria Strain

The bacterial strain *B. cereus* (CECT 131) was obtained from the Spanish Type Culture Collection, University of Valencia. *B. cereus* is rod-shaped, with a size between 1.0–1.2 µm by 3.0–3.5 µm; it is motile, endospore-forming, aerobe to facultative and, being a gram-positive bacteria, it originates 3–8 mm large greyish colonies, often with irregular borders [29]. The bacteria growth period is circa 28 h. The culture media, previously sterilized during 30 min, at 121 °C, had the following composition: 5 g/L beef extract (Sigma-Aldrich, Toluca, Mexico; analytical grade), 10 g/L peptone (Oxoid, Hampshire, UK; analytical grade) and 5 g/L NaCl (PanReac, Darmstadt, Germany, 99.5%). The optimal growth pH was adjusted to 7.2.

### 3.3. Effect of the Different REE on the Bacteria Growth

REE impact on the *B. cereus* growth was assessed using a sterilized culture medium with 20 mg/L of each REE for single and multi-component assays. These Erlenmeyer flaks were inoculated with the bacteria and placed in an orbital shaker (150 rpm) at 30 °C during 30 h. Samples were taken at different moments and centrifuged (Eppendorf MiniSpin, Hamburg, Germany) at 11,000 rpm for 10 min. The resuspended pellet was used to quantify the biomass concentration by optical density (wavelength of 620 nm), while the supernatant was used to quantify the concentration of the REE by Induced Coupled Plasma–Optical Emission Spectrometry, ICP-OES (Optima 8000, PerkinElmer, Shelton, CT, USA). A blank was used as a control, containing medium and bacteria in optimal conditions of growth to correlate with the standard growth of *B. cereus*.

In order the assess the impact that different earth rare metals, in singular and multi-component solutions, may exert on the morphogenesis and on the growth of *B. cereus*, several samples from all the experiments conducted with this bacterium were collected over time, dried, subjected to Gram staining, and subsequently observed under the microscope (Olympus BX51, Midland, Canada).

### 3.4. Zeolite Characterization

The molecular sieve zeolite 13X was purchased from Acros Organics with a particle size from 4 to 8 mesh. This material was evaluated using Scanning Electron Microscopy/Energy Dispersive X-Ray Spectroscopy (SEM/EDS, Phenom-World BV, The Netherlands), determination of pH of zero point of charge (pHzpc) and textural characterization by N_2_ adsorption.

The samples were characterized using a desktop scanning electron microscope (SEM) coupled with energy-dispersive X-ray spectroscopy (EDS) analysis (Phenom ProX with EDS Phenom-World BV, Netherlands). All data were acquired using the ProSuite software (Phenom-World BV, The Netherlands) integrated with Phenom Element Identification software (Phenom-World), used for the quantification of the concentration of the elements present in the samples, expressed in either weight or atomic concentration. The samples of zeolite were placed into aluminum pin stubs with electrically conductive carbon adhesive tape (PELCO Tabs, Manchester, NH, USA). Samples were processed without coating. The aluminum pin stub was then placed inside a Phenom Sample Holder (SR), and different points were analyzed for elemental composition. EDS analyses were conducted at 15 kV with intensity map.

The pHzpc values for zeolite were measured: a solution of 0.01 M NaCl was prepared, previously bubbled with nitrogen in order to stabilize the pH by preventing the dissolution of CO_2_ and the pH was adjusted to different values (1 to 10) by adding diluted H_2_SO_4_ or NaOH. For each pH value, the adsorbent (0.10 g) was added to 25 mL of NaCl. All the flasks were sealed to avoid contact with air and left under moderate agitation (110 rpm) at 25 °C ± 1 °C, for 24 h. The samples were then filtered, using 0.2 μm nylon filters and the pH of the final filtrate was measured and plotted against the initial pH value. The pH at which the curve crosses the line pHinitial = pHfinal is taken as pHzpc.

The textural characterization of the adsorbents was based on the adsorption of N_2_ at 77 K using a Micromeritics ASAP 2010 apparatus, performed at University of Coimbra (Coimbra, Portugal).

### 3.5. Sorption Experiments in Batch Systems

The batch assays were performed using the previously mentioned REE with the zeolite and with *B. cereus* in suspension or in supported biofilm. These assays were all conducted using acetate buffer 0.1 M pH 5.0. The kinetics of the processes were assessed, and the dynamic equilibria were established.

#### 3.5.1. Adsorption by Zeolite

Sorption assays were performed in Erlenmeyers, with zeolite and all REE, in individual and in mixed solutions. These assays were carried out at room temperature (25 ± 1 °C) in rotary shakers at 120 rpm. The zeolite concentration was 5 g/L, while the concentration of each REE was 20 mg/L, for the kinetics evaluation. All kinetics assays were conducted in triplicate, and the results are an average of all the assays.

The equilibria were determined using a fixed concentration of REE, with the zeolite concentration ranging from 0.25 to 10 g/L. These assays were carried out at room temperature at 120 rpm in orbital shakers for 48 h for Tb, Y and Pr, 60 h for La and Ce and 72 h for Eu, the pre-established time periods needed for sample extraction determined by the kinetic assays. Samples for the quantification of metal concentration in the liquid phase by ICP-OES were taken at the beginning, t_0_, and at the end of the assay, t, and the pH was measured. All equilibrium assays were conducted in triplicate and data are an average of all the assays.

#### 3.5.2. Biosorption by Bacteria in Suspension

*B. cereus* was grown for 24 h, at 30 °C, in liquid medium with the composition described in 2.3, in a volume of 0.3 L, and then 0.15 L was transferred to 2 new culture media, and these cultures were grown for 24 h at 30 °C and 120 rpm. After this period, the biomass obtained was centrifuged in sterilized falcons on an Allegra X-15 R centrifuge (Beckman Coulter, Watertown, MA, USA) at 5000 rpm for 15 min at 30 °C. The biomass pellets were re-suspended in a smaller volume of acetate buffer, 0.1 M, with the final concentration stablished to be 11 g_biomass_/L. Then, singular sorbate and mixed sorbate experiments were conducted with the biomass previously concentrated, in order to evaluate the sorption capacity of bacteria towards the REE—La, Ce, Y, Tb, Pr, Eu—individually and in mixed solutions. The experiments were performed for a fixed REE concentration of 20 mg/L, pH 5.00 and a biomass concentration of 0.5 g/L. The Erlenmeyer flasks were placed in an orbital shaker (120 rpm) at room temperature and, at different time intervals, samples were collected and centrifuged (Eppendorf MiniSpin) at 11,000 rpm for 10 min, after which the supernatant was analyzed by ICP-OES in order to determinate the metal concentration. The biosorption assays were conducted in duplicate. 

The equilibrium assays were accomplished with all REE mixed with concentrations ranging from 5 mg/L to 25 mg/L. The assay was performed in an orbital shaker (120 rpm) at room temperature. Previous experiments indicated that the required time for equilibrium to be reached is 48 h. Samples of 4 mL were collected at 0 and at 48 h, and the supernatant was analyzed by ICP-OES.

#### 3.5.3. Sorption by Supported Biofilm

The sorption experiments with zeolite supported B. cereus biofilm started with the growth of the bacteria biomass at 30 °C and 120 rpm, for 18 h, in 10 mL of a liquid medium with the composition mentioned in 2.3. Then, this inoculated medium was transferred to 12 mL of a half-diluted culture medium with 2.5 mg of zeolite. This culture was grown approximately for 72 h at 30 °C and 120 rpm, using a diluted media in order to promote the formation of a robust biofilm and the aggregation of bacterial biomass onto the zeolite. 

Following this period, the supported biofilm was used in sorption kinetics assays, with mixture of all REE. These experiments were conducted in a rotary shaker (120 rpm, Sheldom Manufacturing, Cornelius, OR, USA) at a pH of 5.00, at room temperature, with a cocktail of REE with a fixed concentration of 20 mg/L of each tested element. Samples of 5 mL were taken periodically and analyzed in ICP-OES. 

Equilibrium assays were performed with REE concentrations in the range of 2.5 mg/L to 25 mg/L for each element when tested in a multi-component assay and 8.3 g/L of supported biofilm. These experiments were performed in an orbital shaker (120 rpm) at room temperature, for around 48 h. An initial sample and a sample taken after 48 h of assay were analyzed for metals concentrations.

### 3.6. Analytical Procedures for Quantification of REE

Each liquid sample to be analyzed at ICP-OES (Optima 8000, PerkinElmer) was filtered through a pore size of 0.22 µm and some drops of nitric acid, HNO_3_, (Fisher, Loughborough, UK, 69%) were added to avoid the alteration of the pKa value. The operating conditions were as follows: RF power: 1300 W, argon plasma flow: 8 L/min, auxiliary gas flow: 0.2 L/min, nebulizer gas flow: 0.5 L/min. The plasma view was axial for La, Ce, Tb and Pr and was radial for Y and Eu. The wavelengths (nm) used for each element were: La—408.672, Ce—413.764, Eu—381.967, Y—371.029, Tb—350.917 and Pr—390.844.

### 3.7. Process Modelling

#### 3.7.1. Kinetics Modelling

Kinetics experiments were performed with zeolite and with *B. cereus* biomass in batch systems in order to determine the equilibrium adsorption time and uptake. The zeolite assays were carried out with a support concentration of 5 g/L and with a REE concentration of 20 mg/L for each tested element, for single- and multi-component assays. Moreover, the biomass concentration for single- and multi-component assays was 11 g_biomass_/L, with the same REE concentrations. Finally, the concentration used in the experiments with biofilm was 8.3 g/L, with a REE concentration as previously reported. All assays were carried out at room temperature and shacked at 120 rpm. The adsorbed amount of REE at time t, q_t_ (mg/g) was calculated by: q_t_ = ((C_0_-C_t_) × V)/m(1)
where C_0_ (mg/L) is the initial concentration in solution of the REE under analysis, C_t_ (mg/L) is the concentration of REE at time t. V (L) is the volume of the REE solution and m (g) is the weight of the adsorbent used. Two kinetic models, the Lagergren pseudo-first order (PFO) and pseudo-second order (PSO) were used to justify the experimental data. The least-squares regression method (Origin Pro 8.0. software, Madrid, Spain) was used to fit these models to the experimental data. 

The pseudo-first order (PFO) and the pseudo-second order (PSO) equations used were: q_t_ = q_e_ [1 + exp(−*k*_1_ × t)](2)
q_t_ = (*k*_2_ × q^2^ × e^t^)/(1+ *k*_2_ × q_e_ × t)(3)
where q_t_ (mg/g) is the concentration of solute present in the solid at time t; q_e_ (mg/g) is the adsorption capacity at equilibrium, i.e., the mass of adsorbate per unit mass of adsorbent at equilibrium; *k*_1_ is a rate constant (min^−1^) and reflects a combination of the rate constants of adsorption *k_a_* and desorption *k_d_*; *k*_2_ (g/(mg × min)) is a complex function related to the initial concentration of solute.

#### 3.7.2. Equilibrium Modelling

Equilibrium experiments in batch systems were performed with zeolite and *B. cereus* biomass, as well as with biofilm, in order to establish the adsorption isotherms and respective uptake. A fixed REE concentration of 20 mg/L and an adsorbent concentration ranging between 0.25–10 g/L were used in assays with the zeolite as sorbent. In assays with suspended biomass, a fixed concentration for biomass of 0.08 g_biomass_/L was used. The concentration of biofilm supported on zeolite was 8.3 g/L, and a REE concentration range of 2.5–25 mg/L was considered. The time needed to attain equilibrium for zeolite and for biomass sorption was previously determined in kinetic assays. The adsorbed REE in equilibrium for each condition, q_e_ (mg/g) was calculated through the following equation, where C_e_ (mg/L) is the concentration of REE in solution at the equilibrium, C_0_ is the initial concentration in solution of the REE under analysis, V (L) is the volume of the solution and m (g) is the weight of the adsorbent used:q_e_ = [(C_0_ − C_e_) × V]/m(4)

It is important to highlight that the experimental data were fitted by the Langmuir (Equation (5)) and by the Freundlich (Equation (6)) models, as these have been reported in the literature to have the best fitting parameters for metal sorption isotherms [7,60]:q_e_ = (q_max_ × *K_L_* × C_e_)/(1 + *K_L_* × C_e_)(5)
qe = *K_F_* × C_e_^1/n^(6)
where q_max_ (mg/g) represents the maximum adsorption capacity, *K_L_* (L/mg) is the equilibrium constant related to the free energy of adsorption, *K_F_* [(mg/g)·(L/mg)^1/n^] is the relative adsorption capacity and n is related to the strength of the sorption, which should have values within the range of 1 to 10 for classification for favorable sorption [40]. Other authors consider that the adsorption is favorable when n > 1 [49].

### 3.8. REE Leaching from Zeolite

Desorption assays were conducted with zeolite used in previous adsorption experiments with HNO_3_ acting as a leaching agent. The acid concentrations used were 0.5 and 0.1 M, with a short period of contact between the zeolite and the leaching agent, 2.5 h. The desorption experiments were performed at room temperature in rotary shakers (120 rpm, Sheldom Manufacturing, Cornelius, OR, USA).

### 3.9. Sorption Experiments in Column

The experiments were performed with two columns, 2.5 cm by 30 cm, which were filled with zeolite up to middle height, corresponding to 21 g of zeolite per column. The column beds were washed with distilled water during 4 h with a flow rate of 18 mL/min.

To create a biofilm on the zeolite, a half diluted medium of B. cereus was prepared and then inoculated, passing through one column with a flow rate of 12 mL/min during 48 h. The other column with zeolite was continuously washed with distilled water with a flow rate of 12 mL/min during 48 h. A mixture of REE with a concentration of 10 mg/L for each REE, passed through each column with a flow rate of 3.0 mL/min for 26 h, without re-circulation. This was performed to compare the removal capacities of the zeolite and of the zeolite-supported biofilm. As the metal solution passes through the columns, samples of the outflow were taken at different time points.

After these assays, the columns were subjected to a desorption process with nitric acid at 0.1 M during 5.5 h, with no recirculation and with a flow rate of 3 mL/min, to evaluate the REE recuperation from the zeolite and from the supported biofilm.

## 4. Conclusions

Zeolite has higher yields of adsorption for La, Ce and Pr, with a recovery over 90%, in both single- and multi-solute batch assays, suggesting a higher affinity for these elements. 80% of removal was achieved for Ce, La and Tb when tested in single-solute batch assays with suspended biomass, with no significant difference when tested with multi-solute solutions. Finally, the results for supported biomass on zeolite are very similar to those with zeolite, with over 80% removal for La, Ce and Pr.

The PSO model showed the best fit for the processes kinetics under the majority of the tested conditions, meaning that the adsorption follows a chemisorption mechanism for single- and multi-solute conditions. The equilibria modelling indicated Langmuir to be the best isotherm model under the tested conditions, suggesting a monolayer adsorption process. 

The desorption in batch from zeolite showed a higher recovery for Eu, Tb and Y for single- and multi-solute solutions, with recoveries of over 60% under all tested conditions. La, Ce and Pr had lower recoveries, of 20%, 30% and 40%, respectively, when tested for multi-solute conditions, due to the higher affinity of these elements for the zeolite.

The column assays revealed good adsorption by the zeolite, with or without biofilm, but when both conditions were compared, the biofilm only increased the REE recovery for a short period of time. Desorption from the column was possible, and resulted in higher yields for zeolite without biofilm, indicating that the biofilm increases the resistance to the REE removal from the zeolite.

## Figures and Tables

**Figure 1 molecules-24-01005-f001:**
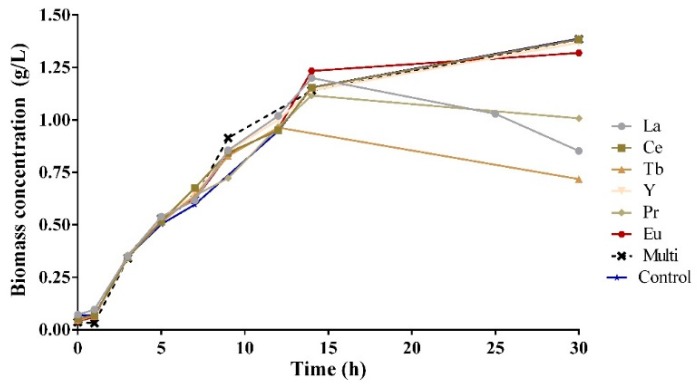
Biomass concentration (g/L) versus time (h), for an initial concentration of 20 mg/L for single REE, multi REE, and control.

**Figure 2 molecules-24-01005-f002:**
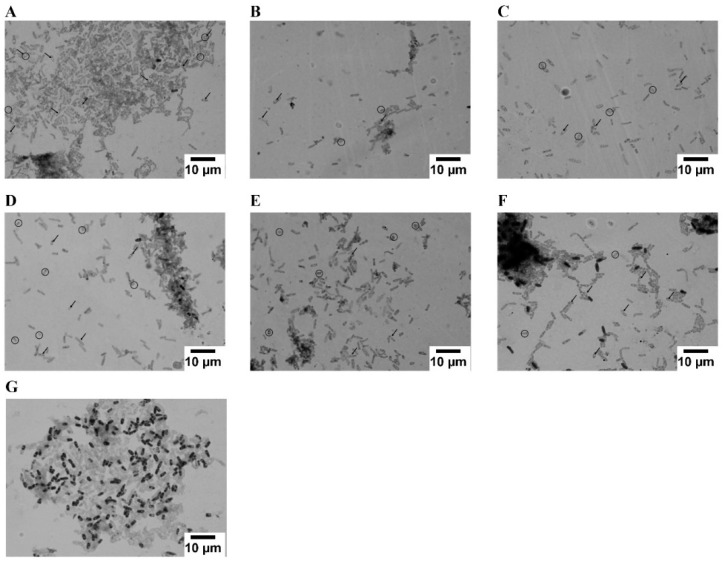
Morphogenesis of *B. cereus* when exposed to 20 mg/L of (**A**) La, (**B**) Ce, (**C**) Y, (**D**) Pr, (**E**) Tb, (**F**) Eu and (**G**) all REE tested. Light-grey-cells—Gram-negative stained cells; Black-cells—Gram positive stained cells. → spores; O—Oval-shaped cells. Objective magnification: 100 X.

**Figure 3 molecules-24-01005-f003:**
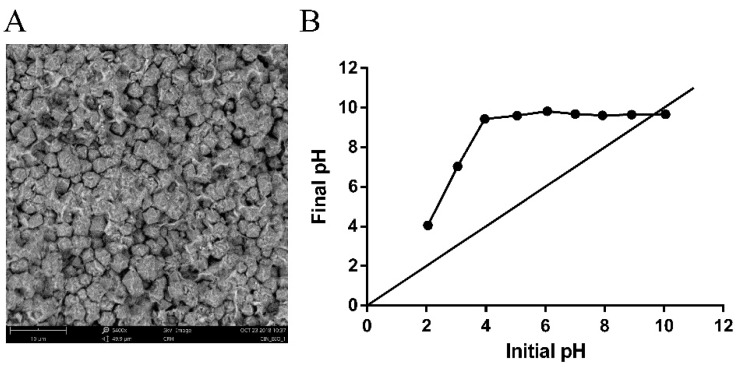
Zeolite characterization by SEM (**A**) and by pH drift method (**B**).

**Figure 4 molecules-24-01005-f004:**
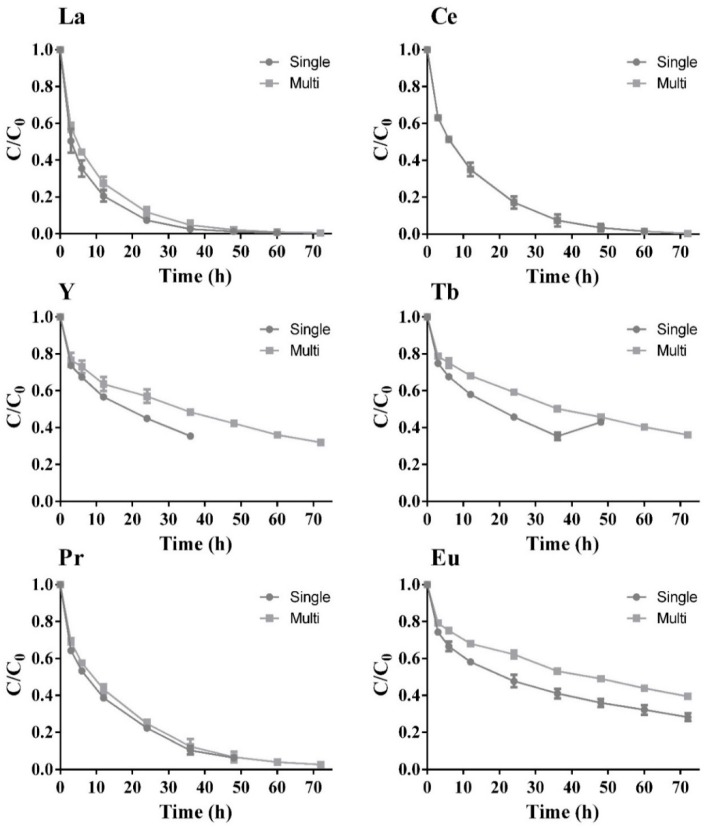
REE normalized concentration, C/C_0,_ over time for zeolite concentration of 5 g/L and REE initial concentration of 20 mg/L for the single- (
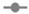
) and multi-component (
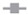
) assays, with the respective standard deviation (*n* = 3, X = 2).

**Figure 5 molecules-24-01005-f005:**
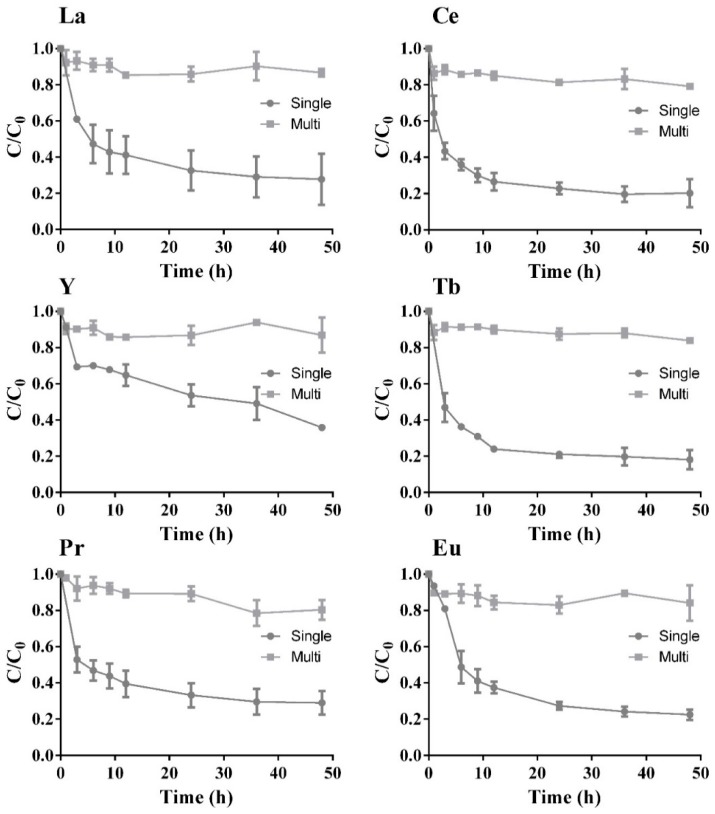
REE normalized concentration, C/C_0_, over time for biomass concentration of 0.5 g/L and REE initial concentration of 20 mg/L for single- (
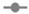
) and multi-component (
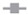
) assays, with the respective standard deviation (*n* = 2, X= 2).

**Figure 6 molecules-24-01005-f006:**
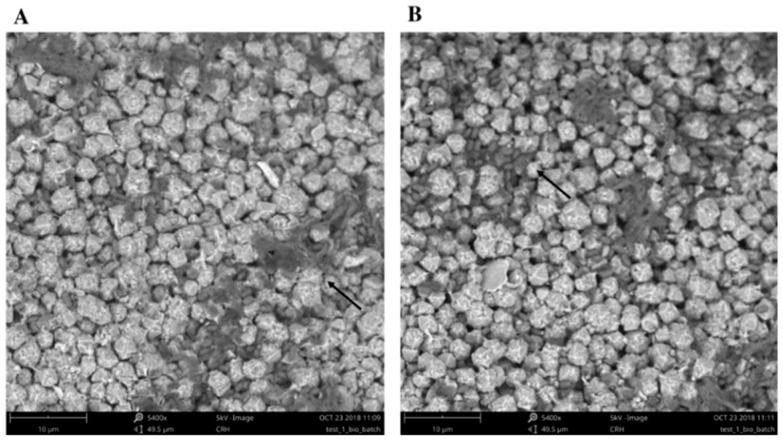
SEM-EDS images of B. cereus biofilm on zeolite. The arrows evidence the presence of biofilm on the surface of the zeolite, perspective 1 (**A**) and perspective 2 (**B**).

**Figure 7 molecules-24-01005-f007:**
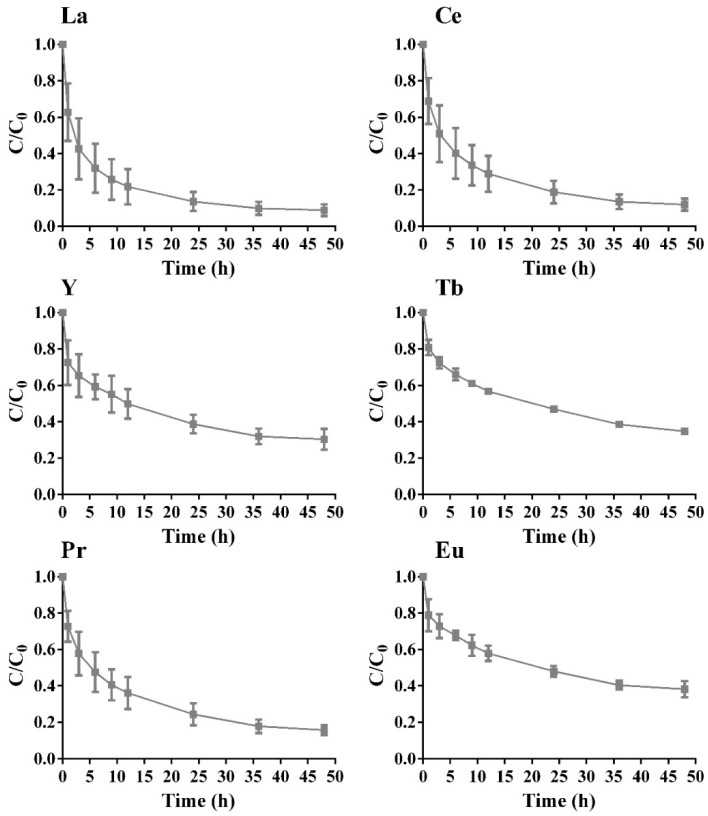
REE normalized concentration, C/C_0_, over time for biofilm supported on zeolite, with a sorbent concentration of 2.5 g/L and initial concentration of 20 mg/L for each REE for multi-component assays, with the respective standard deviation (*n* = 2, X = 2).

**Figure 8 molecules-24-01005-f008:**
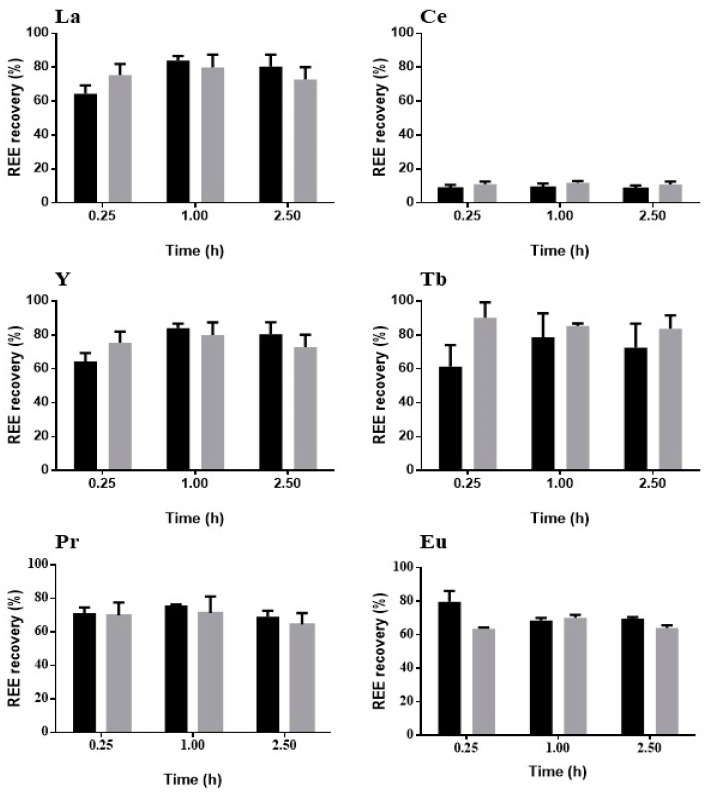
REE recovery from zeolite using HNO_3_ at two different concentrations, 0.1 M (
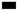
) and 0.5 M (
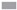
) as leaching agent, using the single-solute adsorption material (*n* = 3, X = 2).

**Figure 9 molecules-24-01005-f009:**
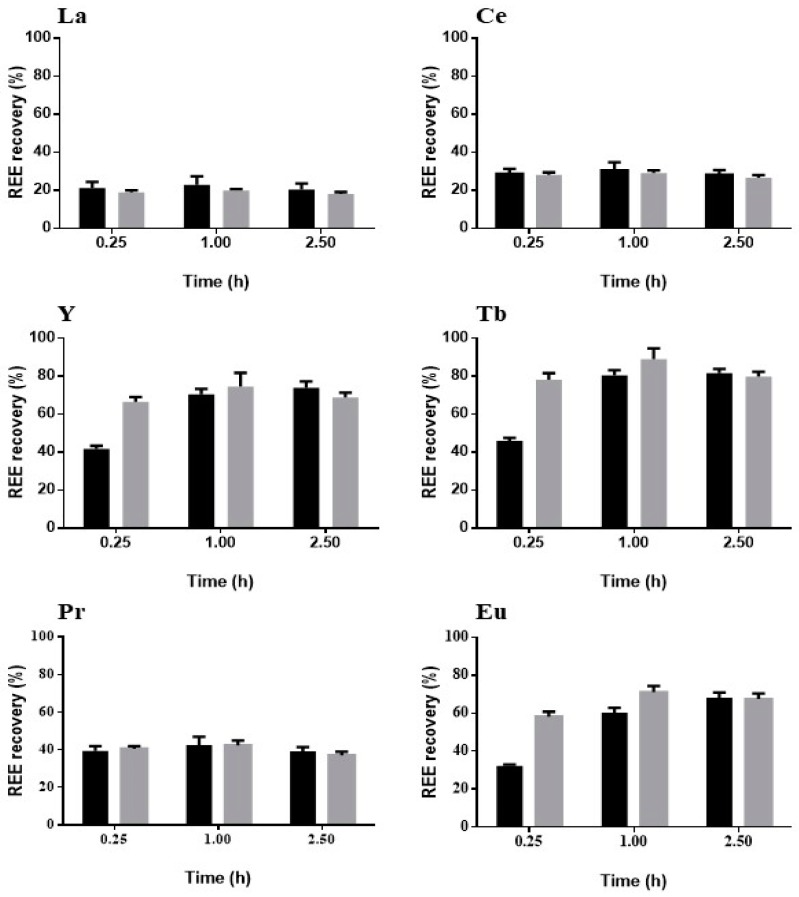
REE recovery from zeolite using HNO_3_ at two different concentrations, 0.1 M (
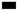
) and 0.5 M (
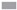
), as leaching agent for the multi-component adsorption material (*n* = 3, X = 2).

**Figure 10 molecules-24-01005-f010:**
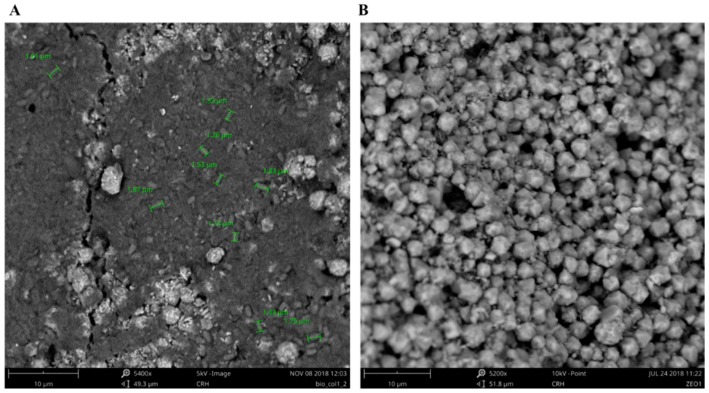
Zeolite covered with biofilm (**A**) and zeolite without biofilm (**B**). Images obtained from SEM.

**Figure 11 molecules-24-01005-f011:**
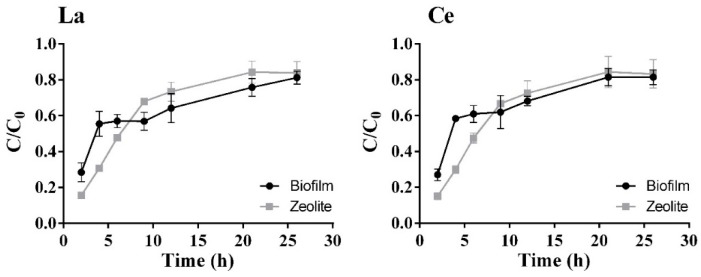
REE C/C_0_ over time with initial total concentration of 10 mg/L for the multiple-component solution. The adsorption of all the elements by zeolite with (

) or without biofilm (

) is presented with the respective standard deviation (*n* = 2, X = 2).

**Figure 12 molecules-24-01005-f012:**
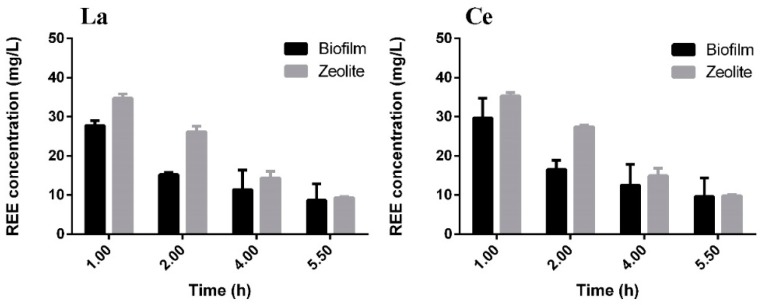
REE concentration when 0.1 M of HNO_3_ was passed through the column after the sorption assays for zeolite with biofilm (
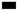
) and without biofilm (
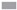
).

**Table 1 molecules-24-01005-t001:** Textural characterization for the zeolite 13X, performed by adsorption of N_2_ at 77 K using a Micromeritics ASAP 2010 apparatus.

Adsorbent	S_BET_ (m^2^/g)	V_total_ (cm^3^/g)	V_micro_ (cm^3^/g)	V_meso_ (cm^3^/g)	Average Pore Size (Å)
Zeolite 13X	576	0.29	0.21	0.08	20

**Table 2 molecules-24-01005-t002:** Uptake values for the multi-component solution and for single-component solutions, in mg/g of zeolite.

	La (mg/g_zeolite_)	Ce (mg/g_zeolite_)	Y (mg/g_zeolite_)	Tb (mg/g_zeolite_)	Pr (mg/g_zeolite_)	Eu (mg/g_zeolite_)
Single solute	3.69	3.87	2.72	3.56	3.31	3.33
Multi-solute	4.00	3.71	3.44	2.27	4.54	2.65

**Table 3 molecules-24-01005-t003:** Atomic and weight percentage for the SEM/EDS analyses of zeolite with individually adsorbed REE.

	La	Ce	Y	Tb	Pr	Eu
Atomic percentage (%)	2.88	1.37	1.38	0.37	2.21	3.39
Weight percentage (%)	17.5	9.36	6.13	3.08	14.2	22.6
Mass adsorbed by zeolite (mg)	32.5	16.6	11.5	9.35	25.3	34.5
Mass of zeolite (mg)	210					
Zeolite and REE proportion (mg zeolite: 1 mg of REE)	6.46:1	12.65:1	18.26:1	22.46:1	8.30:1	6.09:1

**Table 4 molecules-24-01005-t004:** Uptake values for the multi-component solution and for individual component solutions, in mg/g_biomass._

	La (mg/g_biomass_)	Ce (mg/g_biomass_)	Y (mg/g_biomass_)	Tb (mg/g_biomass_)	Pr (mg/g_biomass_)	Eu (mg/g_biomass_)	REE Total (mg/g_biomass_)
Single solute	29.00	33.04	20.93	35.17	35.81	37.75	
Multi-solute	4.53	5.52	0.84	8.60	13.29	6.01	38.79

**Table 5 molecules-24-01005-t005:** Uptake values for the multi-component solution in contact with biofilm supported on zeolite, in mg/g_sorbent_.

	La (mg/g_sorbent_)	Ce (mg/g_sorbent_)	Y (mg/g_sorbent_)	Tb (mg/g_sorbent_)	Pr (mg/g_sorbent_)	Eu (mg/g_sorbent_)
Multi-solutes	2.28	2.12	2.06	1.40	1.84	1.39

**Table 6 molecules-24-01005-t006:** Fitting parameters, square errors and AIC for PFO and PSO for the adsorption kinetics with the different adsorbents (Ads): zeolite (Z), suspended biomass (SB), for the single solute assays.

REE	Ads	PFO				PSO			
		q_e_	k_1_	R^2^	AIC	q_e_	k_2_	R^2^	AIC
**La**	Z	3.62 ± 0.08	0.215 ± 0.021	0.984	−22.50	3.95 ± 0.04	0.080 ± 0.005	0.998	−39.60
	SB	12.12 ± 1.34	0.055 ± 0.021	0.952	5.87	17.45 ± 3.30	0.002 ± 0.001	0.942	7.59
Ce	Z	3.75 ± 0.12	0.125 ± 0.015	0.977	−18.60	4.30 ± 0.09	0.036 ± 0.004	0.995	−30.50
	SB	2.119 ± 0.06	0.497 ± 0.077	0.970	−29.50	2.301 ± 0.02	0.318 ± 0.020	0.998	−53.40
Y	Z	2.55 ± 0.18	0.130 ± 0.028	0.952	−10.70	3.08 ± 0.22	0.047 ± 0.013	0.978	−15.50
	SB	25.42 ± 1.25	0.125 ± 0.019	0.959	17.30	29.86 ± 1.76	0.005 ± 0.001	0.971	14.40
Tb	Z	2.17 ± 0.15	0.132 ± 0.028	0.952	−12.50	2.62 ± 0.18	0.056 ± 0.016	0.979	−17.50
	SB	49.00 ± 0.65	0.013 ± 0.001	0.996	6.52	45.35 ± 1.71	0.421 ± 0.074	0.958	25.90
Pr	Z	3.15 ± 0.14	0.128 ± 0.020	0.966	−14.40	3.70 ± 0.13	0.041 ± 0.007	0.990	−22.80
	SB	45.98 ± 1.80	0.292 ± 0.053	0.952	26.40	50.24 ± 1.21	0.009 ± 0.001	0.990	14.20
Eu	Z	3.01 ± 0.13	0.115 ± 0.021	0.942	−18.20	3.44 ± 0.12	0.042 ± 0.008	0.981	−28.30
	SB	44.77 ± 1.34	0.225 ± 0.026	0.949	22.10	50.23 ± 0.75	0.006 ± 4.6 × 10^−4^	0.997	4.87

q_e_—adsorption capacity at equilibrium calculated from the fitting (mg/g); k_1_—affinity constant of the pseudo-first order model (min^−1^); k_2_—affinity constant of the pseudo-second order model (g·mg^−1^·min^−1^); R^2^—coefficient of correlation; AIC—Akaike Information Criterion.

**Table 7 molecules-24-01005-t007:** Fitting parameters, square errors and AIC for PFO and PSO for the adsorption kinetics with the different adsorbents (Ads)—zeolite (Z), suspended biomass (SB) and supported biomass on zeolite (SBZ)—for the multi-solute assays.

REE	Ads		PFO			PSO			
		q_e_	*k* _1_	R^2^	AIC	q_e_	*k* _2_	R^2^	AIC
**La**	Z	3.94 ± 0.08	0.158 ± 0.015	0.983	−25.00	4.35 ± 0.05	0.050 ± 0.004	0.997	−39.50
	SB	3.90 ± 0.39	1.147 ± 0.678	0.790	5.20	4.26 ± 0.44	0.268 ± 0.207	0.852	3.10
	SBZ	1.94 ± 0.08	0.359 ± 1.940	0.946	− 27.0	2.13 ± 0.04	0.247± 0.028	0.992	−44.30
Ce	Z	3.63 ± 0.09	0.123 ± 0.013	0.978	−23.90	4.10 ± 0.07	0.039 ± 0.004	0.995	−36.70
	SB	4.55 ± 0.36	1.589 ± 0.832	0.863	4.00	4.75 ± 0.38	0.524 ± 0.426	0.890	2.70
	SBZ	1.74 ± 0.08	0.254 ± 0.047	0.940	−27.70	1.93 ± 0.05	0.195 ± 0.030	0.986	−41.20
Y	Z	3.10 ± 0.20	0.077 ± 0.018	0.911	−13.90	3.64 ± 0.25	0.026 ± 0.008	0.954	−19.80
	SB	3.68 ± 0.20	1.009 ± 0.315	0.914	−5.38	3.87 ± 0.18	0.442 ± 0.179	0.951	−9.30
	SBZ	1.68 ± 0,13	0.182 ± 0.047	0.861	−21.60	1.85 ± 0.12	0.155 ± 0.053	0.931	−27.80
Tb	Z	2.10 ± 0.13	0.070 ± 0.015	0.930	−23.10	2.50 ± 0.16	0.033 ± 0.009	0.963	−28.80
	SB	7.25 ± 0.82	0.199 ± 0.079	0.828	10.10	8.27 ± 0.93	0.031 ± 0.017	0.906	6.50
	SBZ	1.37 ± 0.09	0.137 ± 0.028	0.911	−29.10	1.57 ± 0.10	0.115 ± 0.033	0.955	−35.30
Pr	Z	4.42 ± 0.13	0.095 ± 0.011	0.976	−19.20	5.13 ± 0.12	0.023 ± 0.003	0.993	−30.40
	SB	15.52 ± 1.29	0.053 ± 0.010	0.974	6.61	21.83 ± 3.18	0.002 ± 8.53 × 10^−4^	0.967	8.20
	SBZ	1.71 ± 0.09	0.196 ± 0.037	0.933	−27.40	1.90 ± 0.07	0.148 ± 0.028	0.978	−37.80
Eu	Z	2.36 ± 0.14	0.090 ± 0.021	0.907	−18.30	2.74 ± 0.17	0.041 ± 0.012	0.955	−24.80
	SB	5.21 ± 0.41	1.138 ± 0.588	0.779	5.55	5.77 ± 0.46	0.202 ± 0.121	0.860	1.90
	SBZ	1.53 ± 0.11	0.142 ± 0.032	0.891	−25.50	1.73 ± 0.12	0.113 ± 0.037	0.939	−30.90

q_e_—adsorption capacity at equilibrium calculated from the fitting (mg/g); *k*_1_—affinity constant of the pseudo-first order model (min^−1^); *k*_2_—affinity constant of the pseudo-second order model (g·mg^−1^·min^−1^); R^2^—coefficient of correlation; AIC—Akaike Information Criterion.

**Table 8 molecules-24-01005-t008:** Langmuir and Freundlich isotherm fittings for single-solute solutions in zeolite.

REE	Langmuir				Freundlich			
	q_max_	*K_L_*	R^2^	AIC	*K_F_*	N	R^2^	AIC
La	5.52 ± 0.11	1.73 ± 0.13	0.994	−32.3	3.54 ± 0.13	5.44 ± 0.79	0.963	−16.0
Ce	5.30 ± 0.11	4.26 ± 0.72	0.984	−26.4	4.22 ± 0.08	9.78 ± 1.19	0.987	−28.8
Y	8.84 ± 1.36	0.18 ± 0.08	0.980	−8.6	2.81 ± 0.81	3.42 ± 1.28	0.977	−7.6
Tb	3.46 ±0.13	0.35 ± 0.05	0.994	−42.4	1.56 ± 0.11	4.36 ± 0.59	0.992	−39.4
Pr	5.09 ± 0.20	0.69 ± 0.09	0.979	−26.6	2.40 ± 0.13	3.59 ± 0.41	0.966	−21.9
Eu	3.64 ± 0.21	0.69 ± 0.27	0.982	−23.8	2.47 ± 0.26	9.66 ± 4.37	0.974	−21.1

q_max_—maximum adsorption capacity (mg/g); *K_L_*—equilibrium constant related to the free energy of adsorption (L/mg); *K_F_*—relative adsorption capacity [(mg/g)·(L/mg)^1/n^]; N—relative strength of the sorption; R^2^—coefficient of correlation; AIC—Akaike Information Criterion.

**Table 9 molecules-24-01005-t009:** Langmuir and Freundlich isotherm fittings for multi-solute solutions in different adsorbents (Ads): zeolite (z), suspended biomass (SB) and supported biomass on zeolite (SBZ).

REE	Ads	Langmuir				Freundlich			
		q_máx_	*K_L_*	R^2^	AIC	*K_F_*	N	R^2^	AIC
**La**	Z	6.04 ± 0.21	1.86 ± 0.211	0.955	−29.10	3.50 ±0.10	4.26 ± 0.30	0.956	−29.50
	SB	6.56 ± 1.68	0.77 ± 0.56	0.869	9.20	2.97 ± 0.46	3.30 ± 0.80	0.954	4.00
	SBZ	5.04 ± 0.57	0.64 ± 0.13	0.989	−23.90	1.85 ± 0.09	1.55 ± 0.17	0.968	−16.30
Ce	Z	4.86 ± 0.18	1.89 ± 0.27	0.948	−32.70	3.04 ± 0.11	4.72 ± 0.56	0.918	−25.90
	SB	5.21 ± 0.38	5.30 ± 3.22	0.907	−0.99	3.82 ± 0.25	6.17 ± 1.42	0.968	−3.71
	SBZ	5.55 ± 0.64	0.33 ± 0.06	0.994	−29.50	1.31 ±0.06	1.42 ± 0.11	0.987	−23.50
Y	Z	5.80 ± 0.27	0.46 ± 0.11	0.984	−27.30	3.07 ± 0.26	5.40 ± 1.05	0.984	−26.90
	SB	4.45 ± 1.90	0.21 ± 0.24	0.840	3.70	1.10 ± 0.57	2.26 ± 1.28	0.851	3.30
	SBZ	10.2 ± 3.19	0.04 ± 0.001	0.995	−34.40	0.41 ± 0.03	1.13 ± 0.06	0.994	−32.40
Tb	Z	2.59 ± 0.11	1.42 ± 0.54	0.980	−45.20	1.98 ± 0.14	12.09 ± 5.59	0.974	−42.00
	SB	11.1 ± 0.62	0.91 ± 0.31	0.972	7.20	5.99 ± 0.23	5.05 ± 0.37	0.997	−4.40
	SBZ	5.07 ± 0.64	0.08 ± 0.01	0.996	−36.20	0.40 ± 0.03	1.32 ± 0.08	0.993	−31.20
Pr	Z	4.36 ± 0.12	0.15 ± 0.16	0.969	−45.40	2.67 ± 0.06	5.05 ± 0.34	0.977	−50.10
	SB	12.0 ± 0.79	1.87 ± 0.99	0.959	10.10	8.20 ± 0.91	7.23 ± 2.63	0.968	8.85
	SBZ	6.06 ± 1.58	0.17 ±0.06	0.984	−24.10	0.86 ± 0.08	1.27 ± 0.14	0.971	−19.80
Eu	Z	2.30 ± 0.09	1.34 ± 0.49	0.984	−54.50	1.76 ± 0.13	12.07 ± 4.98	0.98	−51.80
	SB	9.62 ± 2.06	0.13 ± 0.06	0.964	2.50	1.47 ± 0.01	1.79 ± 0.01	0.988	22.10
	SBZ	9.01 ± 2.16	0.03 ± 0.009	0.997	−39.90	0.29 ± 0.02	1.14 ± 0.05	0.997	−38.00

q_max_—maximum adsorption capacity (mg/g); *K_L_*—equilibrium constant related to the free energy of adsorption (L/mg); *K_F_*—relative adsorption capacity [(mg/g)·(L/mg)^1/n^]; N—relative strength of the sorption; R^2^—coefficient of correlation; AIC—Akaike Information Criterion.

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
