# Peer review of "Recovery of Rare Earth Elements from Wastewater Towards a Circular Economy"

_molecules, 2019, doi:10.3390/molecules24061005_

Reviewer 1 Report

Molecule: Manuscript number 455633

Recovery of rare earth elements from wastewater towards a circular economy

General comments

The authors provided the necessary background of the rare earth elements (REE), with sufficient literature citation and addressed the reason for the need of new technology for the extraction of rare earth elements from the simulated waste.  In this study, the authors also investigated possible way of rare earth element (REE) removal via adsorption using zeolite, and recovery options. In addition, the toxicity of some of REE were evaluated and included. Overall, the authors, the methodology is well designed, conducted relevant experimental works, and interpreted well the analysis results. This article could be published, after the authors take into consideration the following points. 

The authors must read the article thoroughly and correct simple grammar to improve the readability of the article.

Keywords: change plural key words to singular

Line80-82: Not readable well

Line 151: correct as: using pH drift method

Line 434- 435: not clear, please revise

Line 547: ICP-OES: can be defined as inductively coupled plasma-optical emission spectrometry

Line 555-556: SEM-EDS is an instrument while the pH of zero point of charge (pHzpc) and textural characterization are not?? Please revise.

Line 589: Why different experimental times used for extraction of REE is not clear. Is there any adsorption time optimisation conducted elsewhere?

Line 664-665: see equation number 4, it is not correct, please rewrite correctly.

Q. All the work conducted and presented clearly have shown possible recovery of REE from the simulated wastewater. However, no environmental samples were collected and checked the possible recovery of REE using the developed methods. Therefore, do you think that similar recovery is possible from the environmental samples which composed of different matrixes or compositions?

Author Response

In response to your review of our paper, we want to thank you for the attention paid to our manuscript and for the comments, which were quite constructive and took all of them into consideration in this new edition of our article. Now, we will address the comments in the following order:

Comment 1: The authors must read the article thoroughly and correct simple grammar to improve the readability of the article.

        Ans: The text was thoroughly reviewed, grammar and spelling mistakes were corrected              and readability was hopefully improved;

Comment 2: Keywords: change plural key words to singular

        Ans: Words were changed as suggested;

Comment 3: Line80-82: Not readable well

        Ans: The whole sentence was removed as it just repeated an idea stated above in the manuscript;

Comment 4: Line 151: correct as: using pH drift method

        Ans: Text changed accordingly;

Comment 5: Line 434- 435: not clear, please revise

        Ans: Test was rewritten as suggested by the reviewer;

Comment 6: Line 547: ICP-OES: can be defined as inductively coupled plasma-optical emission spectrometry

        Ans: Text changed accordingly;

Comment 7: Line 555-556: SEM-EDS is an instrument while the pH of zero point of charge (pHzpc) and textural characterization are not?? Please revise.

        Ans: The reviewer is quite right, all of them are instrumental methodologies, text is rewritten.

Comment 8: Line 589: Why different experimental times used for extraction of REE is not clear. Is there any adsorption time optimisation conducted elsewhere?

        Ans: Those values were selected from kinetics assays data. Some explanation was added to the text.

Comment 9: Line 664-665: see equation number 4, it is not correct, please rewrite correctly

        Ans: Indeed, equation was corrected and some explanation added to the text.

Comment 10: All the work conducted and presented clearly have shown possible recovery of REE from the simulated wastewater. However, no environmental samples were collected and checked the possible recovery of REE using the developed methods. Therefore, do you think that similar recovery is possible from the environmental samples which composed of different matrixes or compositions?

        Ans: It is our intention to use the developed method to real industrial and mining effluents and to environmental samples in a very near future, although based on the state of the art, other usual components of such effluents tend not to disturb the equilibria and the kinetics of REE removal. Nevertheless, this is a task to be performed within the current goals of this research project. Due to its relevance, this issue was referred to in Introduction.

Hoping that these changes may improve the quality of our work, for which we are very thankful, we submit to your appreciation a new version of our article.

Reviewer 2 Report

The authors studied a technology for the recovery of rare earth elements from synthetic wastewater. In this research two alternative technologies, adsorption with zeolite and adsorption with zeolite and bacteria were compared. The manuscript is well presented, the results are justified by analytical techniques and modeling elaborations have also been carried out.

The following is a list of minor remarks:

Introduction could be further expanded considering also similar research on real wastewater from industries.

The authors in rows 80-82 explained that this process aims to enhance also the re-use of REE present in wastewater. In the whole manuscript, the technology to be used for recovery from the solutions is not discussed, but the research stops at the desorption treatment. In order to separate REE one from the other, different studies on selective REE recoveries are present in the literature (eg. Innocenzi et al. 2018 “Application of solvent extraction operation to recover rare earths from fluorescent lamps” Journal of Cleaner Production)

Figure 8 and 9, check the correspondence of the bars with the correct nitric acid concentration, probably is present a mistake, otherwise the discussion in row 452-454 is not congruent with what is shown by the figure.

From the desorption treatment authors obtained very low values of cerium recovery, the resin used allowed to test higher nitric acid concentration or to increase the contact time in order to obtain higher recoveries?

In Discussion section, I recommend inserting some comments on the possibility of regenerating the resin. Many studies are reported in the literature.

Author Response

In response to your review of our paper, we want to thank you for the attention paid to our manuscript and for the comments, which were quite constructive and took all of them into consideration in this new edition of our article. Now, we will address the comments in the following order:

Comment 1: Introduction could be further expanded considering also similar research on real wastewater from industries.

        Ans: It is our intention to use the developed method to real industrial and mining effluents and to environmental samples in a very near future, although based on the state of the art, other usual components of such effluents tend not to disturb the equilibria and the kinetics of REE removal. Nevertheless, this is a task to be performed within the current goals of this research project. Due to its relevance, this issue was referred to in Introduction;

Comment 2: The authors in rows 80-82 explained that this process aims to enhance also the re-use of REE present in wastewater. In the whole manuscript, the technology to be used for recovery from the solutions is not discussed, but the research stops at the desorption treatment. In order to separate REE one from the other, different studies on selective REE recoveries are present in the literature (eg. Innocenzi et al. 2018 “Application of solvent extraction operation to recover rare earths from fluorescent lamps” Journal of Cleaner Production)

        Ans: In fact there are two main lines of research. The one indicated by the reviewer, with solvent extraction of REE from the leachate, eventually followed by selective precipitation for their recycling, and another that may foresee the catalytic application of the REE, once the elements are already entrapped within the zeolite, which is our next step in this research project. The REE are being tested in environmental catalysis in oxidation of VOC in mild conditions. Some text was added to the manuscript as suggested by the reviewer;

Comment 3: Figure 8 and 9, check the correspondence of the bars with the correct nitric acid concentration, probably is present a mistake, otherwise the discussion in row 452-454 is not congruent with what is shown by the figure.

        Ans: The reviewer is right, the captions were wrong and now they have been corrected.

Comment 4: From the desorption treatment authors obtained very low values of cerium recovery, the resin used allowed to test higher nitric acid concentration or to increase the contact time in order to obtain higher recoveries?

        Ans: Probably the zeolite would stand higher concentrations or longer contact times, but the assays were only performed in a comparative basis to demonstrate that leaching is possible, to differentiate the specificities of each element and to distinguish the elements more prone to be leached from the support. Next step of this research will be the definition of tailored protocols for the individual recover for each element.

Comment 5: In Discussion section, I recommend inserting some comments on the possibility of regenerating the resin. Many studies are reported in the literature.

        Ans: Reference to the eventual regeneration of the zeolite was added to the text as  suggested, much in line with previous works published by our team.

Hoping that these changes may improve the quality of our work, for which we are very thankful, we submit to your appreciation a new version of our article.
